# Structure, Intensity and Player Duels in Under-13 Football Training in Switzerland

**DOI:** 10.3390/ijerph17228351

**Published:** 2020-11-11

**Authors:** Jonas Uebersax, Ralf Roth, Tobias Bächle, Oliver Faude

**Affiliations:** Department of Sport, Exercise and Health, University of Basel, 4051 Basel, Switzerland; jonas.uebersax@bluewin.ch (J.U.); ralf.roth@unibas.ch (R.R.); tobias.baechle@baspo.admin.ch (T.B.)

**Keywords:** soccer, youth, children, health, training programming

## Abstract

We evaluated the structure (i.e., the different training parts), contents (i.e., the various activities used), intensity and occurrence of contact situations and headers during training sessions in under-13 football in Switzerland. A total of 242 players from 20 different teams on average aged 11.4 (SD 0.7) years participated. The participants were filmed during a typical training session while they were equipped with a heart rate sensor. The sessions were systematically recorded to allow for detailed analyses. Furthermore, a preliminary and explorative analysis of the influence of the level of play on these results was conducted. The overall findings indicated that training included 33.4% playing forms, 29.5% training forms, 28.4% inactivity time and 8.7% athletics. The highest heart rates were achieved in the playing forms (166 min^−1^, 83% HR_max_) compared to the other two activities (training forms 154 min^−1^, 77% HR_max_; athletics 150 min^−1^, 75% HR_max_). Each player had 12.8 duels and 0.6 headers per training. Overall, most duels were conducted from the anterior direction. Playing forms induce higher cardio-circulatory load as well as a better learning environment. Potentially dangerous situations like contact with other players or headers occurred in a single player on average every six min during a training session.

## 1. Introduction

Regular physical activity is important for the development of children [1,2]. Football (soccer) is the most popular sport in the world, especially among children. In Switzerland, almost 30% of all boys and 17% of all girls aged 10 to 14 years play football [3]. Football clubs are frequently the first points of contact with the organized sports system. It has been shown that children who play club football in their leisure time exhibit a higher overall daily physical activity [4]. Organized football is, therefore, a promising opportunity to get a large number of children physically active. Research in non-performance-oriented children’s football and, thus, evidence-based measures on how to structure and conduct football training are rare [5].

The important aspects of football training are how to structure a training session and what content to include. An Australian video-based study has shown that coaches frequently used individual and drill forms during the warm-up period, according to the Football Federation Australia (FFA) curriculum for under-11 to under-13 age groups (skill acquisition phase). The learning period (the main part of the session) consisted mainly of small-sided games and the final part almost exclusively of a training game [6]. Drill forms have predetermined movement sequences, predicted results and little decision-making ability [6]. Playing forms, on the other hand, have a play-like character with open decisions and constantly changing conditions, thus including tactical and technical elements simultaneously. It is assumed that children enjoy playing more than drills, that they typically learn implicitly, differentially and undermine a more effective development that supports superior performance [7,8,9]. According to contemporary research and theory, the emphasis in children’s and youth football should be on playing forms [7]. In addition, a study on talent identification states that perception-related cognitive and decisive abilities and game performances are important foundations for the selection of promising players [10]. Certain coaches’ qualifications are required in order to design training sessions at the appropriate level in terms of structure, learning behavior, teaching and intensity. In children’s football in Switzerland, many coaches are insufficiently or even not at all trained. Data on the typical structuring of training sessions in children’s football in Switzerland or other Central European countries is currently lacking.

Football is also a suitable means to improve fitness and health in children. The developing organism is exposed to important stimuli targeting particularly the cardiovascular and the musculoskeletal systems as well as metabolism [11,12,13]. Various studies showed that playing football leads to increased physical fitness and considerable health benefits. Therefore, football, especially conducted with small-sided games, appears to be a highly effective form of training for children [12,14,15,16]. In order to achieve physiological adaptations, it is important that the intensity, frequently assessed by heart rate measurements, is sufficiently high. For instance, Bendiksen et al. [17] examined school children aged eight to nine years during football sessions (small-sided games, three vs. three). The boys achieved an average heart rate (HR) of 158 (SD18) min^−1,^ corresponding to 78 (8)% of maximal heart rate. Compared to other sports, playing football usually achieves high heart rate values (based on average HR and proportion of total time at intensities greater than 80% of maximal HR). However, data on training intensity in children´s football is still sparse.

Playing football is, unfortunately, also associated with the risk of sustaining an injury [18]. Football is among those team sports with the highest injury risk in boys and girls [19]. In general, injuries can have far-reaching and long-term consequences in terms of physical inactivity, arthrosis, obesity and a limitation of future activities [20]. More than three-quarters of all injuries are located in the lower limbs [21]. The incidence of injuries in young football players increases with increasing age. The majority of injuries occur in high-intensity contact situations with high biomechanical loads. Contact injuries are a particular problem, with nearly 60% of all injuries in children aged from 7 to 12 years resulting from contact with an opponent, the ball or floor [21]. It is, therefore, important to evaluate contact situations in children´s football. Although a minor part of all injuries (5 to 10%) [18,21], head injuries have gained considerable media attention during the last years. In particular, acute traumatic head injuries (including concussions) and the possibility of long-term damage to the brain through repetitive microtrauma induced by frequent heading are currently subject to intense discussion [22,23,24]. This has led to a ban of headers in under-11 football and a call to avoid it in under-13 football in the United States. Current knowledge on the occurrence of duels and headers during training sessions in children’s football is insufficient.

Based on the above-mentioned rationales, our study aimed at evaluating (i) the structure and contents, (ii) the intensity and (iii) the occurrence of contact situations and headers in training sessions in under-13 football in Switzerland by video analyses and heart rate assessment. A further aim was a preliminary and exploratory analysis of the influence of the level of play on these outcomes.

## 2. Methods

### 2.1. Participants

A total of 242 children (5 girls) from 20 different teams took part in the study. The children were, on average, 11.4 (SD 0.7) years old, weighed 41.5 (SD 8.0) kg and were 150.5 (SD 7.9) cm tall. All children played in the under-13 age category and were officially licensed to their football clubs. The study protocol is in accordance with the ethical criteria outlined in the Declaration of Helsinki. The regional ethics committee (Ethikkommission Nordwest und Zentralschweiz) declared that the study is in line with the national guidelines for human research (Req-2020-00639). All players, as well as their parents, signed a written informed consent after having been informed about the study.

In this age group, the number of players per team in official matches in Switzerland is nine. The teams were chosen so that each region of the association, northwestern Switzerland, was covered. According to this, both urban and rural regions were involved. Most of these teams trained twice a week. All teams had coaches who have completed at least the lowest coaching license (beginners’ course) in Switzerland. However, an additional certificate (license category C) is officially required for the age group studied. Exactly three-quarters of the coaches had this or even a higher-level license (category UEFA B). All four regional playing levels of this age group were included in this study.

The Promotion league is the most performance-oriented class in regional recreational football at any youth level. Eight teams played on this level. The remaining teams were from the lower categories, first league (*n* = 8), second league (*n* = 3) and third league (*n* = 1). We combined the three lower, recreational leagues (*n* = 148 players) for statistical comparison with the performance-oriented Promotion league (*n* = 94 players).

### 2.2. General Design

The training sessions took place on the grounds of the respective clubs. Each club was visited once, resulting in a total of 20 analyzed training sessions. The measurements were carried out within 2.5 months (September to November 2018) and took place in all weather conditions. The coaches were informed about the visit. Before the session, anthropometric data of all players were measured. In addition, all players were fitted with a chest strap to measure their heart rate. The coach was asked about his coaching license. On the pitch, we installed a camera in order to record all activities of the players. During the training session, players and coaches performed the training as planned for this day.

The investigation aimed to assess the general situation in children’s football. Thus, the training sessions were not discussed beforehand, nor were the coaches provided with any possible classification or evaluation criteria. This led to a large variety of training sessions performed.

### 2.3. Procedures

#### Video Recordings

The training sessions were recorded on video. A digital camera (iPhone 8, Apple, Silicon Valley, CA, USA; resolution 3840 × 2160, 8.3 megapixels; frame rate 60 fps) was fixed on a tripod. We paid attention to all players being visible on the video. Therefore, we placed the camera outside the field and, if the facility allowed it, on an elevated ground. The entire training session was captured. Three video recordings were interrupted for short time periods due to technical reasons. During these periods, the training contents were noted by hand. This notation could be used for training structure analysis, but not for investigating the duels.

### 2.4. Session Analysis

A training session was considered started when the coach started the session verbally, mostly in a group. Previous activities on the field or setting up material were not counted as training time. The coach verbally announced the end of the training after the last training part.

The trainings were analyzed in two ways (for detailed definitions, see Table 1). First, we evaluated the sessions regarding the training activities, according to O’Connor et al. [6]. The three main activities were playing forms, training forms and inactive time. In addition to these activities, we added an athletic part (only athletic activities were performed) as a separate category. In parallel to this classification, structural parts of the training were surveyed based on the educational curriculum of the Swiss Football Federation and the curriculum of Football Federation Australia [25]. The following classification was used: Warm up; Learning and Teaching - main part; Training game; Finishing part. The finishing part was added as the youth coaches in Switzerland are educated according to a certain training structure, where this training part is integrated. During the finishing part, teams should do some exercises for fun or for emotional regulation after the final training game for the last couple of minutes of the session. There can be shots, challenges, small competitions, cool down or anything else. This part is not a main part of training but was nevertheless considered as training time. The coaches in Switzerland are taught the “GAG” model (G1 = Training theme-related playing form (10–15 min), A = Analytical part (tactical/technical focus; 30 min), G2 = controlled game related to the topic of the session or free game (20–30 min)), which is supplemented by a warm-up in advance and a small finishing part at the end of the training. The differentiation between the warm-up and the main part was not always obvious. One criterion was that the first themed playing form completes the warm-up as coaches are correspondingly educated in the “GAG” teaching method.

In general, the analyses in this part refer to the team as a whole and not to the individual player. If a particular training form was performed, the time for this exercise was measured until the end of the exercise. Even if some players had waiting times, these were not evaluated separately. Particularly in the case of athletic parts, breaks within the exercises are unavoidable. As soon as the first player had started an activity, it was considered as started. As soon as the last player had completed his activity, this part was considered finished. In most instances, the coach indicated this by a verbal signal.

### 2.5. Heart Rate Measurement

The heart rate measurements were performed with a Polar Team^2^ system (Polar Electro Oy, Kempele, Finland) [27]. A belt equipped with electrodes and a transmitter was fixed around the chest. The recorded data were transmitted directly from the transmitter to a base station. The base station was connected to the PC and provided real-time data on the PC screen. The recording of the data was manually started and stopped. Data were downloaded after the training session, and downloaded data were used for further analysis. In order to estimate the maximum heart rate, we used the following formula: 208 − 0.7 × age. This formula has been shown to result in an accurate estimate of HR_max_ for children [28]. For the evaluation of the different training activities and parts, the times in the respective periods were evaluated. During playing forms, only the heart rates of the field players were taken into account. With regard to the athletic activities, it must be mentioned that different contents were used for improving the physical fitness of the children and mostly combined (speed, endurance, strength, agility). This led to a variety of loads, which cannot be allocated to a specific athletic area and, thus, were analyzed in combination. We started heart rate recordings manually. This process was recorded with the camera and used for synchronizing video recordings and heart rate measurements.

### 2.6. Duels and Headers

Duels and headers were evaluated by visual inspection. We defined a duel as a situation with one player trying to conquer the ball from his opponent by tackling or blocking. We further differentiated the direction from which the (first) contact came anterior (+/−45° deviation from the sagittal plane in viewing direction), lateral (+/−45° deviation from the frontal plane in both lateral directions), and posterior (+/−45° deviation from sagittal plane opposing to viewing direction). Additionally, we assessed running duels as situations with two players running side-by-side in the same direction, fighting for the ball over a longer running distance (>5 m), either with one player controlling the ball or with both players trying to reach a ball first. We classified air duels as situations with at least one player jumping vertically and one opponent nearby (<1 m), also trying to reach the ball or disturbing the opponent. Finally, every ball that was intentionally touched with the head was counted as a header. A subsample (*N* = 3 teams) was analyzed by two raters independently in order to estimate the accuracy of definitions. We observed a discrepancy between raters for a total number of duels of 2% (anterior 2%, lateral 12%, posterior 10%), for running duels of 14%, for air duels of 14% and for headers of 2.5%, respectively.

### 2.7. Data Analysis

Data are presented mainly descriptive and mostly in minutes, percentages of total training time or in absolute numbers (duels and headers). Time data are given as means with standard deviation (SD) and as minimum and maximum. Absolute numbers are presented as medians with upper and lower quartiles and as minimum and maximum. Mean differences between teams of the different performance levels were calculated together with bootstrapped 95% confidence limits based on 5000 resamples [29]. Two-sided *p* values were calculated by the Mann–Whitney test for the differences between the different performance levels and by the Wilcoxon test for difference in heart rates during different training activities.

## 3. Results

### 3.1. Training Activities

The total average training time was 81.4 (SD 5.8) minutes (ranging from 69.0 to 90.0 min). The duration of the different training activities is presented in Table 2. Playing forms represent about one-third of a training session, with training forms lasting slightly shorter. Inactivity time also constituted nearly 30% to total training time. Athletic contents represent less than 10% of total training time. We found a difference in the time that was invested in playing and training forms between teams of the Promotion league compared to the lower levels. Promotion level teams conducted on average 7.9 min (95% CI 1.4, 14.3, *p* = 0.04) more playing forms and 6.8 min (−0.5, 14.0, *p* = 0.15) less training forms compared to the lower level teams (Figure 1). Inactive time and athletic contents were similar between levels (*p* > 0.5).

### 3.2. Training Structure

With regard to the different training parts, we observed that the main part constituted more than half of total training time, with the warm-up and training game making up about 20% of total training time each (Table 3). The main part was 9.4 min (−2.3, 21.1, *p* = 0.11) longer, and the training game was 7.9 min (−0.4, 16.1, *p* = 0.09) shorter in the Promotion teams compared to the lower level teams (Figure 2). Warm-up and the finishing part were similar in both playing levels (*p* > 0.35).

### 3.3. Heart Rate

Average team heart rates during the different training activities are presented in Table 4. HR was highest during playing forms and lowest during athletic activities. The mean difference between playing forms and athletics was 15 min^−1^ (95% CI 8, 24, *p* = 0.006), between playing forms and training forms 12 min^−1^ (95% CI 8, 16, *p* < 0.001) and between training forms and athletics 2 min^−1^ (95% CI −6, 11, *p* = 0.6; Figure 3). There were no obvious differences between playing levels (*p* > 0.3). Players spent 34.4% of total training time at a HR below 70% of HR_max_, 38.5% between 70 and 85% of HR_max_ and 27.1% above 85% of HR_max_.

### 3.4. Duels and Headers

The number of duels and headers is presented in Table 5. On average, 38.2% of the total training time consisted of situations where duels can occur. A player was involved in nearly 13 duels per training, corresponding to one duel every 6.4 min on average. More than two-thirds of all duels were conducted from the anterior direction. The overall number of headers per player and training session was less than one. Promotion league teams conducted more headers (1.0 headers per player per session) than lower-level teams (0.4 headers per player per session). One team from the Promotion league completed specific header training resulting in a total of 209 headers in this particular training session.

## 4. Discussion

The aims of this study were to evaluate the structure and contents, the intensity as well as contact situations and headers in training sessions in under-13 football in Switzerland. To the best of our knowledge, this is the first study in a Central European country on this topic. We found that playing forms amounted to the largest part of training time. This observation was more pronounced in the higher-level teams. Athletic contents represented less than 10% and inactivity time nearly one-third of the total training time. The average heart rate was highest during playing forms, reaching an average of nearly 83% of the maximal heart rate. On average, each player performed more than 12 duels and less than one header during one training session, i.e., a potentially dangerous situation every 6 min.

The average training duration in our study was very similar to previous studies, i.e., around 80 min [6,26]. In our study, fewer playing forms on average (33.4% vs. 35.6–40.9%) and training forms inside the lower range (29.5% vs. 22.3–64.4%) were applied compared to other studies [6,7]. The inactive time (28.4% vs. 36.8%) was lower in our study than in the one by O’Connor et al. [6]. It is important to note that we added an athletic part. This section accounted for less than 10% of the total training time and was partly included in the subcategory of training forms in former studies. The athletic part consisted of individual athletic exercises, and we intended to separate it from the training form part to get a more accurate analysis. Athletic exercises were conducted without the ball and with the intention to improve physical abilities rather than football-specific skills. We observed that no team included any specific prevention program in the training sessions, although it has been shown that such programs can result in relevant performance improvements [30] and, more importantly, can reduce injury incidence by about 40% already in these age groups [31,32]. According to the long-term athletic development model, improving the general physical and motor abilities of children by appropriate exercises can be regarded as the main goal of youth training in any sports in order to promote children’s health and physical fitness, to maximize physical activity participation rates, to reduce the risk of injury and to ensure long-term health and well-being [33]. According to our data, this aim seems not appropriately addressed by current practices in children’s football in Switzerland.

The largest difference between the performance levels (Promotion vs. lower level) was that the higher level teams applied more playing forms (38.4% vs. 30.1%), while the lower level teams spend more time in training forms (33.6% vs. 24.1%). Likewise, at the lower level, the main part did almost not include any playing forms and teams performed mainly one open (final) game at the end of the training. The Promotion league teams were closer to the recommended training standards in Switzerland, as they included a sufficient amount of playing forms in the main part and the whole training session. The curricular recommendation is that playing forms should represent at least 30 min. These differences are unlikely due to the coaches’ qualifications, as we found no relevant differences in the obtained diplomas between the higher and lower level coaches. Five coaches did not even have a sufficient diploma for this age category. The time spent in the different training phases (warm-up/main part/final game) did not differ relevantly compared to the results of O’Connor et al. [6] and fits with the recommendation of the Swiss Football Association. The only difference in this study was the finishing part, which was introduced by the Swiss Football Association and is suggested to complete every training session. We observed that this objective has not yet arrived at the regional pitches. As actually intended by the Swiss Football Association, at least a section of the main part should be a thematic playing form. Our results show that almost half of the surveyed teams did not play any form of the game in the main section, and many only had the final game as a form close to the real game in their training sessions. So-called small-sided games are mostly missing in the main part, and the players are offered too little time for implicit, game-related learning in which they must make their own decisions and undergo a more effective learning process [9,34].

Heart rate data show that playing forms are an appropriate way to induce beneficial adaptations as children reached the highest heart rate levels while playing (>80% of estimated maximum). Sperlich et al. [35] mentioned different heart rate zones in order to comprehensively describe fitness training in under-14 football players: below 60% HR_max_, 60 to 70% HR_max_, 70 to 80% HR_max_, 80–90% HR_max_, above 90% HR_max_. Our results were in the middle to upper range of this classification system. The playing forms reached the range from 80 to 90% of HR_max_. Heart rate is usually used as an indicator of training intensity. Improvements in peak oxygen uptake are only achievable with a sufficient training stimulus [36]. Baquet et al. [37] showed that relevant aerobic training effects occur at intensities between 78% and 95% of HR_max_ in adolescents. Also, Obert et al. [38] reported improvements in maximal oxygen uptake in boys and girls with endurance training at 80% HR_max_ and above. Faude et al. [14] observed large improvements in 20 m shuttle-run performance after a 6-month football training at 78% HR_max_ in overweight children. In summary, heart rates during playing forms in our study were in intensity zones, which have been shown to induce relevant improvements in children and adolescents.

The investigation of the frequency of duels in recreational football for players under 13 years of age is a novelty of our study and can serve as a basic record of the current situation in typical football training sessions in under-13 players. Contact injuries to children between the ages of 7 and 12, including an opponent, account for a large proportion of injuries [21]. On average, the coaches organized 38.2% of the training time with exercises, which bear the risk of contact between opponents. The direction of a duel is particularly interesting because players may be differently prepared for a duel coming from anterior vs. posterior direction. Most duels in our study occurred in an anterior direction. In anterior duels, both players can anticipate the following action, possibly limiting the risk of injury, whereas, in lateral and posterior duels, at least one player frequently is not aware of the approaching situation. In this regard, it is of particular interest that Faude et al. [23] observed that, indeed, most injuries to the head occurred due to an impact to the face or the forehead, whereas the majority of concussions resulted from an impact to the occiput. The authors speculated that players were mostly unaware of the impact to the occiput and, therefore, could not prepare for the impact. Running duels and air duels were relatively rare in our study. Most air duels occurred after throw-ins.

Headers are a hotly debated topic in youth and children’s football [22,23,24]. Attention must be paid to the children’s brain and skull because both structures are still developing. It has been shown that after a single training session, including 6 to 20 headers, amateur players aged 15 to 25 years show cognitive changes [39,40]. It is discussed whether such acute effects potentially resulting from repetitive micro traumata due to frequent heading may result in long-term neurocognitive sequelae [23]. Repeated head impacts can, in the long run, lead to structural brain changes, which have been observed in former football players [41]. Recently, Mackay et al. [42] found that mortality from neurodegenerative disease was higher, and dementia medication was prescribed more frequently in several thousand former Scottish football players compared to matched controls. Although the latter findings were obtained in former professionals, the current situation in recreational club football is also of particular importance as the vast majority of active footballers are playing on an amateur level. In our study, each player conducted less than one header per training session. This is in line with training data of a recent large-scale observational study in eight European countries (median 4 to 6 headers per training session with large variability in under-10 and under-12 teams) with heading incidence during match play being even lower [43,44]. Altogether, these findings point towards a low danger of heading-related risks at this level of play. As heading incidence in match play seems to be lower, there is the opportunity to further reduce the frequency of headers due to the currently unclear short and long-term effects [45], particularly as heading frequency increases with increasing age [43,44].

## 5. Limitations

Some limitations need to be addressed. For instance, we did not take the quality of training contents as well as the quality of the coaches’ instructions into account. In addition, our study is a cross-sectional study with a limited number of teams from the northwestern part of Switzerland. However, we included teams from rural as well as from urban areas and teams of different performance levels. It remains unclear whether our results can be generalized to other European countries or countries in other parts of the World. Furthermore, training intensity was evaluated by heart rate measurements. Additional assessment of subjective ratings of perceived exertion could have given further insight into the experienced intensity of the training session. The assessment of duel situations was conducted by visual inspection and, thus, is prone to subjective bias. Furthermore, we used only one camera. Though we paid attention to all players being visible on the video, we cannot exclude that some relevant contact situations were not correctly detected.

## 6. Conclusions

In summary, our findings show that playing forms, on average, account for about one-third of a typical training session in under-13 football, with relevant differences between performance levels. Only those teams playing at a higher level achieved the recommendation of the Swiss Football Association for playing forms. We also found that playing forms led to the highest cardio-circulatory intensity and, thus, are appropriately high to induce physiological adaptations. In addition, our findings show that the frequency of duels and headers is generally low.

### Practical Applications

Based on these findings, the following recommendations for football practice can be made: In contrast to current recommendations, playing forms are inadequately represented and should be applied more frequently. Playing forms not only allow for more effective development of technical and tactical performance but also of cardio-circulatory capacity. Athletic training parts focusing on general, neuromuscular development and, therefore, particularly supporting the prevention of injuries, may also become a substantial part of training sessions. Coaches should be aware of potentially risky contact situations in children’s football.

## Figures and Tables

**Figure 1 ijerph-17-08351-f001:**
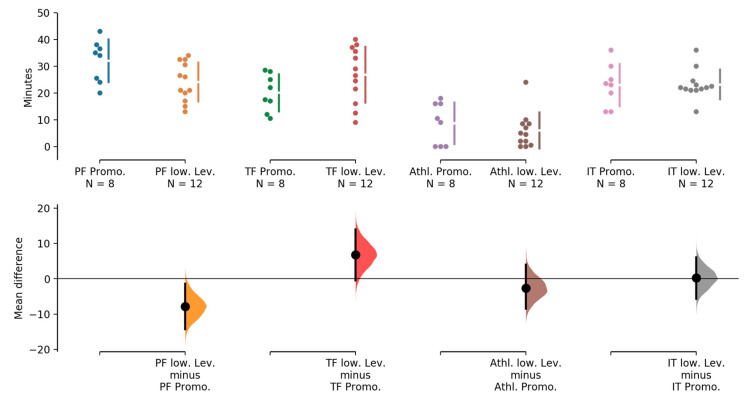
Comparison of training activities between the two performance levels (promotion vs. lower level). (PF = playing form; TF = training form; Athl. = athletic; IT = inactive time; Promo. = promotion; low. Lev. = lower level).

**Figure 2 ijerph-17-08351-f002:**
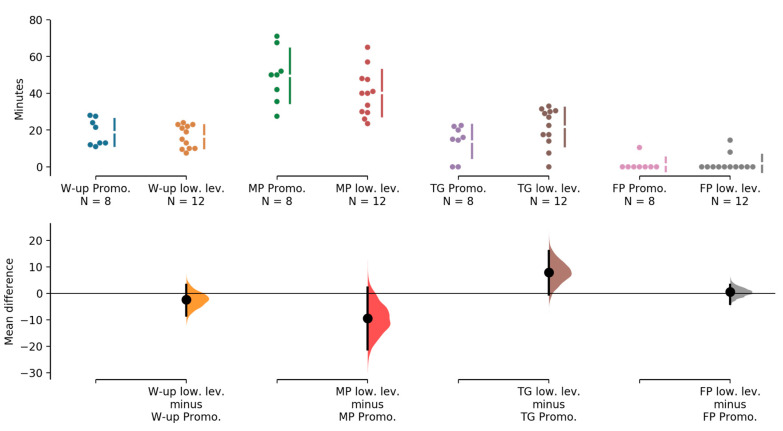
Comparison of training parts between the two performance levels (promotion vs. lower level). (W-up = warm-up; MP = main part; TG = training game; FP = finishing part; Promo. = promotion; low. Lev. = lower level).

**Figure 3 ijerph-17-08351-f003:**
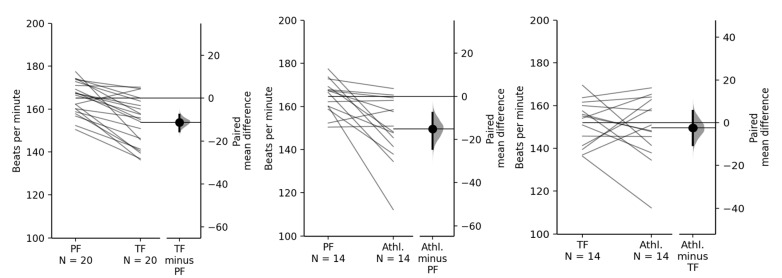
Mean heart rates of each team for different training activities. (PF = playing form; TF = training form; Athl. = athletic. *N* = 14 teams for the comparison regarding the athletic part, as 6 teams did not perform athletic exercises).

**Table 1 ijerph-17-08351-t001:** Definitions of training activities and parts. Adopted, modified and complemented from [6,25,26].

Training Activities	
**Playing Form**	Activities that have an emphasis on match or gameplay, including: **Small-sided activities** Game activities with small teams (2 to 4 players) **Larger activities** Game activities with teams of 5 or more players **Phase of play** Unidirectional match-play towards one goal.
**Training Form**	Activities that have a predicted outcome or isolated skill component, including: **Individual** Activities with the player working on skills by himself **Paired** Activities with players working with one other individual **Drills** Activities during training where the player is performing predetermined actions or movements with a set sequence and minimal alternative options.
**Athletics**	Activities with players working without a ball in order to improve specific athletic skills
**Inactive Time**	Moments with neither activity in athletics, training or playing forms including: **Freeze in position** Coach stops the activity to talk to the players, and the players remain in their current position **Player huddle** Coach stops the activity, and players to coming together for a discussion or explanation **Drink break** Coach stops the activity for having a short rest and drink **Transition** Transitioning between activities or inactivity periods.
**Training Parts**	
**Warm-up**	The first activity of the session, which introduces the core skill and is the only component of the session where drill-type exercises should be used. The warm-up ends with the first playing form. When started with a playing form, the duration of the first form is considered as the warm-up period. Running, strengthening, and other conditioning exercises are counted as a warm-up as long as they are used for this purpose.
**Learning and Teaching–Main Part**	The part where conscious teaching and learning takes place. The coach designs an activity, which provides repetition of realistic game scenarios, coupled with task-based coaching using effective interventions (i.e., quality feedback and questions). This part is delimited from the last part by the fact that there is still much coaching and the games are not yet too similar to the traditional form. This also includes conditioning exercises, which are clearly separated from the warm-up.
**(Final) Training Game**	A finishing game with many elements of the real game being present, but organized to promote the skill of the coaching session.
**Finishing Part**	Training part that takes place after the actual final game. It is for fun or as a cool down.

**Table 2 ijerph-17-08351-t002:** Mean duration of training activities during training sessions. Data presented as means with standard deviation and minimum and maximum.

Training Activity	Overall		Promotion		Lower Level	
	Mean (SD)	Min–Max	Mean (SD)	Min–Max	Mean (SD)	Min–Max
Playing form (min)	27.2 (5.8)	13.0–43.0	32.0 (7.9)	20.0–43.0	24.1 (7.3)	13.0–34.0
(% of total training time)	33.4 (7.1)	16.0–52.8	38.4 (9.5)	24.0–51.6	30.1 (9.1)	16.2–42.4
Training form (min)	24.0 (9.9)	7.0–40.0	20.1 (6.9)	10.5–28.5	26.9 (10.4)	9.0–40.0
(% of total training time)	29.5 (12.2)	8.6–49.1	24.1 (8.3)	12.6–34.2	33.6 (13.0)	11.2–49.9
Athletic (min)	7.1 (7.1)	0.0–24.0	8.7 (7.8)	0.0–18.8	6.0 (6.7)	0.0–24.0
(% of total training time)	8.7 (8.7)	0.0–29.5	10.4 (9.4)	0.0–22.6	7.5 (8.4)	0.0–30.0
Inactive time (min)	23.1 (6.4)	13.0–36.0	22.9 (7.8)	13.0–36.0	23.2 (5.5)	13.0–36.0
(% of total training time)	28.4 (7.9)	16.0–44.2	27.5 (9.4)	15.6–43.2	29.0 (6.9)	16.2–44.9

**Table 3 ijerph-17-08351-t003:** Mean duration of training parts during training sessions. Data presented as means with standard deviation and minimum and maximum.

Training Part	Overall		Promotion		Lower Level	
	Mean (SD)	Min–Max	Mean (SD)	Min–Max	Mean (SD)	Min–Max
Warm-up (min)	17.4 (6.6)	7.5–28.0	18.8 (7.3)	11.0–28.0	16.4 (6.2)	7.5–24.0
(% of total training time)	21.4 (8.1)	9.2–34.4	22.5 (8.7)	13.2–33.6	20.5 (7.7)	9.4–30.0
Main part (min)	43.8 (13.9)	23.5–71.0	49.4 (14.8)	27.5–71.0	40.1 (12.6)	23.5–65.0
(% of total training time)	53.8 (17.1)	28.9–87.2	59.3 (17.8)	33.0–85.2	50.1 (15.7)	29.3–81.1
Training game (min)	18.5 (10.5)	0.0–33.0	13.8 (9.0)	0.0–22.5	21.7 (10.5)	0.0–33.0
(% of total training time)	22.7 (12.9)	0.0–40.5	16.6 (10.8)	0.0–27.0	27.1 (13.1)	0.0–41.2
Finishing part (min)	1.7 (4.2)	0.0–14.5	1.3 (3.7)	0.0–10.5	1.9 (4.6)	0.0–14.5
(% of total training time)	2.1 (5.2)	0.0–17.8	1.6 (4.4)	0.0–12.6	2.4 (5.7)	0.0–18.1

**Table 4 ijerph-17-08351-t004:** Mean heart rates in different training activities during training sessions. Data presented as means with standard deviation and minimum and maximum.

Heart Rate in Training Activity	Overall		Promotion		Lower Level	
	Mean (SD)	Min–Max	Mean (SD)	Min–Max	Mean (SD)	Min–Max
Heart rate in playing form (bpm)	165 (8)	150–176	164 (7)	152–175	167 (8)	150–176
(% of estimated HR_max_)	82.9 (3.9)	75.0–88.0	81.8 (3.6)	76.0–87.5	83.5 (4.0)	75.0–88.0
Heart rate in training form (bpm)	154 (11)	136–170	153 (10)	140–168	155 (13)	136–170
(% of estimated HR_max_)	77.0 (5.7)	68.0–85.0	76.6 (5.0)	70.0–84.0	77.3 (6.4)	68.0–85.0
Heart rate in athletics (bpm)	150 (15)	112–168	153 (12)	134–165	148 (17)	112–168
(% of estimated HR_max_)	75.0 (7.7)	56.0–84.0	76.5 (6.1)	67.0–82.5	74.1 (8.7)	56.0–84.0

**Table 5 ijerph-17-08351-t005:** Duels and headers during training sessions. Data presented as medians with interquartile range and minimum and maximum.

Type of Contact	Overall		Promotion		Lower Level	
	Median (IQR)	Min–Max	Median (IQR)	Min–Max	Median (IQR)	Min–Max
Anterior duel	71 (40)	28–170	78.5 (46.5)	28–101	61 (29)	41–170
Anterior duel per player	6.1 (3.1)	2.8–12.1	6.4 (3.2)	2.8–7.9	6.0 (3)	3.0–12.1
Lateral duel	29 (17)	17–82	30 (12)	24–63	29 (19.5)	17–82
Lateral duel per player	2.5 (1.3)	1.1–5.9	2.3 (1.7)	2.0–4.8	3.0 (0.9)	1.1–5.9
Posterior duel	21 (13)	6–54	18.5 (10)	6–29	22 (13)	6–54
Posterior duel per player	1.6 (0.9)	0.5–3.9	1.6 (0.4)	0.7–2.2	1.6 (1.2)	0.5–3.9
Running duel	16 (12)	0–38	16 (10)	0–38	13 (13.5)	0–32
Running duel per player	1.3 (0.9)	0.0–2.5	1.3 (0.4)	0.0–2.5	1.3 (0.9)	0.0–2.2
Air duel	1 (3)	0–23	2.5 (11.5)	1–23	1 (3)	0–21
Air duel per player	0.1 (0.2)	0.0–2.3	0.2 (0.8)	0.1–2.3	0.1 (0.2)	0.0–2.3
Header	7 (8)	0–209	14 (36.5)	4–209	4 (5.5)	0–24
Header per player	0.6 (0.8)	0.0–14.9	1.0 (3.5)	0.3–14.9	0.4 (0.5)	0.0–2.7
Total duels	127 (84)	91–328	143 (88)	91–215	125 (76.5)	93–328
Total duels per player	12.8 (5.1)	6.1–23.4	12.7 (4.6)	8.5–16.1	12.8 (5.0)	6.1–23.4

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
