# Peer review of "Structure, Intensity and Player Duels in Under-13 Football Training in Switzerland"

_ijerph, 2020, doi:10.3390/ijerph17228351_

Round 1

Reviewer 1 Report

I would like to congratulate the authors for their work. I think that is has been a hard work, but, mainly in the orientation of the discussion and practical applications they should make some changes. Due to some reasons (see limitation), I think that the article should focus more discussion and conclusions sections to those variables extracted from video analyses. I encourage the authors to make some changes, and then I will be willing to make another revision.

INTRODUCTION

• Although I think that the introduction provides sufficient background and include all relevant references, maybe, once the suggestions are addressed, it can be improved.

METHOD Video recordings

• Since it has been published a survey about electronic performance and tracking systems, specially, for the optimal use of radio frequency systems (Rico-González et al., 2020), it has become in a reference article to report information about the use of electronic performance and tracking systems, specially, radio-frequency systems. Although in the present study no radio frequency device was used, the authors should consider any criteria classified as a “general criteria” to perform an optimal description of the used camera system. For example, the frame capacity and the used frames for data recording. • It could be interesting an inter-researcher agreement in video analysis. Session analysis

• I think that Table 1 present all tasks types and their definition, which is suitable and well presents. It is only an opinion, but, although it is clear, maybe is too large and it has several information that it is well known, so, in few times a lecturer will read it. Hence, could be interesting an additional figure that explain the main training structure, or change this figure by the table, in order to allow, with a unique view, the main idea? For example, I should use LUCIDCHART (free programme) to perform figures. Heart rate measurement • Since it has been shown that real-time data could be less accurate than those data downloaded after the training session (see reference: (Aughey & Falloon, 2010)), I think that it is necessary, at least, the citation of an article that assess the validity of the used Polar Team2 system (Polar Electro Oy, Kempele, Finland) in real time.

DISCUSSION

• I think that an assessment of the quality of training sessions is interesting to show to the coaches some limitations about their planification. However, I think that it is highly interesting the comparison between the data recorded and the mean results of these data extracted from U13 matches. In other words, have the players perform a similar demand than in a match? It could be interesting because although the recorded training sessions lacks a certain specific task such as injury prevention programmes (as you reported), the other injury risk found could be a large difference between training and match demands. So, although in this article matches have not been analyzed, it could be interesting a comparison between analyzed training sessions and U13 matches in the published literature. It is because large differences could be the main lack in these teams´ training sessions. Once it has been addressed, it is suitable a more specific analysis about more specific lacks.

LIMITATIONS

• Since the technology has developed, the manufacturers have allowed more than 200 variables extracted from them. I agreed with the authors that intensity is an important parameter to be measured. In fact, soccer players have shown an evolutionary progression performing the accelerations/decelerations, that contemporary competitive match play requires (Harper et al., 2019). In soccer, players make around 220 high intensity efforts (Harper et al., 2019; Sarmento et al., 2014), or accelerate over 178 m during a competition match, while decelerating over 162 m (Akenhead et al., 2013). However, heart rate to analyses training intensity is a way, but, it is not common, at least, since the variables such as accelerations/decelerations or distance covered at different intensities are available. In fact, Casamichana et at. (2019) looked for the principal components to analyze soccer, and heart rate was not between the variables that perform these principal components. So, the analysis of training intensity only though heart rate is a limitation, and the analysis though video camera makes the study interesting.

CONCLUSIONS

• Considering aforementioned limitation, and the main analysis was made through the video camera, I don't think you've established a robust rationale for these analyses (duels, headers). How would coaches and/or sport scientists use this information when working with youth players? Please make a wide practical application about it.

REFERENCES

Akenhead, R., Hayes, P. R., Thompson, K. G., & French, D. (2013). Diminutions of acceleration and deceleration output during professional football match play. Journal of Science and Medicine in Sport, 16(6), 556-561. https://doi.org/10.1016/j.jsams.2012.12.005

Aughey, R. J., & Falloon, C. (2010). Real-time versus post-game GPS data in team sports. Journal of Science and Medicine in Sport, 13(3), 348-349. https://doi.org/10.1016/j.jsams.2009.01.006

Casamichana, D., Castellano, J., Gómez Díaz, A., & Martín-García, A. (2019). Looking for Complementary Intensity Variables in Different Training Games in Football: Journal of Strength and Conditioning Research, 1. https://doi.org/10.1519/JSC.0000000000003025

Harper, D. J., Carling, C., & Kiely, J. (2019). High-Intensity Acceleration and Deceleration Demands in Elite Team Sports Competitive Match Play: A Systematic Review and Meta-Analysis of Observational Studies. Sports Medicine, 49(12), 1923-1947. https://doi.org/10.1007/s40279-019-01170-1

Rico-González, M., Arcos, A. L., Rojas-Valverde, D., Clemente, F. M., & Pino-Ortega, J. (2020). A Survey to Assess the Quality of the Data Obtained by Radio-Frequency Technologies and Microelectromechanical Systems to Measure External Workload and Collective Behavior Variables in Team Sports. Sensors, 16.

Sarmento, H., Marcelino, R., Anguera, M. T., CampaniÇo, J., Matos, N., & LeitÃo, J. C. (2014). Match analysis in football: A systematic review. Journal of Sports Sciences, 32(20), 1831-1843. https://doi.org/10.1080/02640414.2014.898852

Reviewer 2 Report

It was possible to conduct the study with the use of more cameras, for example three, wchich would give a better picture of the whole.

It is worth noting that the article confirms the low level of training in the field of physical fitness in children training football. Under the age of 13, training should be focused on shaping the motor base, not game.

Round 2

Reviewer 1 Report

I would like to congratulate the authors for their work, and for addressing my comments, although one of them remains incomplete.

I would like to see a practical application about the conclusions of the article and practical applications. For example, the authors said: “We also found that playing forms led to the highest cardio-circulatory intensity and, thus, are appropriately high to induce physiological adaptations”. However, since the other type of training forms are suitable for a holistic training process, the authors could make an overall proposal about the implementation of all training forms during the training process. And, on the other hand, what is the practical application for: “our findings show that the frequency of duels and headers is generally low”.

In addition, I would like to add a section under the name “what this article adds?” or something like that, in which the authors explain the most relevant information extracted from this research. Explain the differences between this article´s proposal and other articles related to the topic.
